# The role of organisational- and country-level factors in the volume and public visibility of business and management research

**Kevin Credit***, **Olga Ryazanova**, **Peter McNamara**

Maynooth University, Maynooth, Co. Kildare, Ireland

* kevin.credit@mu.ie.

**Data Availability Statement:** Data used in this paper all come from publically-available sources or using publically-available methods (described in

## Abstract

Using a multilevel modelling approach to analyse a novel dataset of academic publications at all business schools in 11 European countries, this paper finds that the influence of organisational- and country-level contextual factors on researchers varies considerably based on the type of institution and the development level of the country they are located in. At the organisational-level, we find that greater spatial connectivity–operationalised through proximity to nearby business schools, rail stations, and airports–is positively related to scientific research volume and public dissemination (news mentions). While this result is significant only for high-income countries (above EU-average 2018 GDP per capita), this is likely because the low-income countries (below EU-average 2018 GDP per capita) examined here lack a 'critical mass' of well-connected universities to generate observable agglomeration effects. At the country-level, the results indicate that in high-income countries, less prestigious schools benefit from higher rates of recent international immigration from any foreign country, providing a direct policy pathway for increasing research output for universities that aren't already well-known enough to attract the most talented researchers. In low-income countries, recent immigration rates are even stronger predictors of research performance across all levels of institutional prestige; more open immigration policies would likely benefit research performance in these countries to an even greater extent. Finally, the paper's results show that, in low-income countries, a composite measure of a country's quality of life (including self-rated life satisfaction, health, working hours, and housing overcrowding) is positively related to research outcomes through its interaction with school prestige. This suggests that the lower a country's quality of life, the more researchers are incentivised to produce higher levels of research output. While this may in part reflect the greater disparities inherent in these countries' economic systems, it is noteworthy–and perhaps concerning–that we have observed a negative correlation between country-level quality of life and research performance in low-income countries, which is particularly felt by researchers at less prestigious institutions.

the paper). The sources include Web of Science, Altmetrics, Times Higher Education, EuroGlobalMap, Eurostat, and the OECD. For the open methodology for collecting research performance indicators from Web of Science, please refer to Ryazanova et al. (2017): https://doi.org/10.1016/j.jwb.2017.09.00.

**Funding:** The author(s) received no specific funding for this work.

**Competing interests:** The authors have declared that no competing interests exist.

## Introduction

What role does location play in predicting academic researchers' performance? Does it matter in which country and institution one engages in scientific enquiry, and if yes, how important the features of academic institutions and societal contexts are? What are the mechanisms through which location might influence research performance? Research production has traditionally been seen in society as an individually-driven pursuit of knowledge. This historical perception is reflected in the structure of the most prestigious and widely known scientific awards, such as the Nobel Prize and Fields Medal, which are all given to individual scholars. While individual abilities and motivation are important, this individualisation of scientific credit fails to recognise the true role that organisations, governments and other meso- and macro-level factors play in supporting knowledge creation and dissemination. The model of science in which talented individuals achieve outstanding results as wholly-independent scholars is no longer viable. Studies in sociology and economics of science has long pointed to the influence of research environment, including reward structures and activity of peers (among other factors), on individual research performance [1–3].

At the macro-level, research on the spatial distribution of scientific output has consistently found a skewed distribution of research performance in favour of the US and UK, followed by other OECD nations, then the rest of the [4–9]. Significant variance in scientific output also exists both across OECD nations and within them. While a variety of explanations have been forwarded to explain this variance—including general levels of economic development, government investment in research, the Anglo-centric development of the modern academic system, institutional prestige, spatial collaboration networks, and agglomeration effects—the comparative importance of each of these features, and how they interact with one another across spatial scales, remains unclear.

These macro-level differences in research performance, university prestige, and enrolment should also be understood in the context of the new geopolitics of higher education, in which states increasingly view higher education funding and policy as an avenue of 'soft' power through which they can foster positive national images, shape global and regional influence, and achieve foreign policy and economic goals [10–12]. Given the important role that universities play in the contemporary globalised knowledge economy, higher education policy has become an increasingly important component of national economic policy and competition [13–15]. At the same time, geopolitical relations and events can influence the development of particular higher education systems, particularly in relation to their ability to attract and retain talented researchers and students.

Thus, understanding what drives scientific performance is important for audiences at micro-, meso-, and macro-levels. At the micro-level, individual researchers making employment and mobility choices often consider whether a potential employer (or a destination of mobility) is going to offer an environment conducive to research activities [16]. At the meso-level, decision-makers in academic institutions seek to formulate strategies supporting research productivity, which is one of their core performance dimensions. At the macro-level, policy-makers try to increase the competitive potential of research and innovation systems within their nations, creating conditions for attracting and retaining the best researchers and better positioning their higher education systems within the global economic hierarchy [11,15]. In particular, in the uncertain economic environment of recent years, in which decision-makers are increasingly using research funding as a core instrument for advancing science, it is beneficial for them to have a sophisticated understanding of the full suite of factors related to scientific production.

To better understand the meso- and macro-level drivers of spatial inequality in research output, this paper uses a novel dataset on research output and public visibility–within the business and management discipline–for universities located in 11 European countries. Specifically, this paper uses a carefully-designed multilevel modelling approach–split between high- and low-income countries (those with GDP per capita above and below the EU average in 2018)–to empirically analyse the comparative importance of the features of academic institutions themselves and their national societal contexts in predicting research outcomes. Then, given the observed results, we further contextualise the relationships by looking at cross-level interactions between the significant features.

The results of the analysis indicate that both organisational- and country-level factors are significantly related to research performance, but that the influence of these factors generally differs between low- and high-income countries. For high-income countries, business school prestige (measured by business school accreditation and university ranking) and spatial connectivity (measured by distance to other business schools, air, and rail service), increase research volume and public visibility, likely through increased knowledge spillovers and the enhanced ability for researchers at well-connected universities to create and maintain scientific collaboration networks. Location in larger media markets (which are the most spatially well-connected regions) may also influence the public dissemination of research at these universities. Usefully, these findings suggest that regional economic development policy and infrastructure investment can positively influence the knowledge output of local academic institutions in these countries. Perhaps even more interestingly, at the country-level we found that higher rates of recent international immigration (age 15–64 from any foreign country) tend to bolster research volume primarily for *less-prestigious* business schools. This points to the value of government policies that ease international mobility in helping to increase the research performance of institutions that are not already world-renowned leaders in a given discipline.

On the other hand, while school prestige remains significant for low-income countries, spatial connectivity is not, and the relationship with immigration works in a different way: in these countries, *more* prestigious schools benefit the most from higher rates of recent immigration form any foreign country (although all universities benefit to some extent). In addition– somewhat counterintuitively–we also found that a composite measure of a low-income country's quality of life (measured through a composite index of self-rated life satisfaction, health, working hours, and housing overcrowding) is positively related to research volume and public visibility through its interaction with school prestige. In countries with lower quality of life ratings, prestige isn't a significant differentiator for research productivity. However, in countries with higher quality of life ratings, researchers at less prestigious schools produce much less than those at more prestigious schools. These findings suggest the strong role of selection in resource-poor environments: high quality human capital (local and immigrant) gravitates towards the most prestigious workplaces–possibly because those offer better salaries and working conditions. It is also possible that in low-income countries with low quality of life, scholars often cannot avoid the necessity of working longer hours, which results in a higher number of publications and news mentions. However, this necessity also has the potential to create work-life balance tensions that should be investigated in more detail in future research. While this finding must be understood in the context of the business and management discipline studied here, it raises an interesting normative question about the existing organisation of scientific practice: if research output for those at less prestigious institutions in low-income countries is negatively related to holistic measures of quality of life, is this the outcome that should ultimately be prioritised?

## Spatial distribution of research performance

### Organisational-level predictors of research performance

The literature has explored a broad range of organisational factors that contribute to research production (measured as the volume of publications) and public visibility of this research (measured as mentions in news sources) [3]. Among those factors, institutional prestige has consistently been shown to have a positive relationship with research performance [1,17]. There are at least two reasons for this relationship. Firstly, prestige enables organisations to attract resources, which can then be used to support research activity. Secondly, institutional prestige is partly driven by research performance, which is included in the methodology of most university rankings. This makes the true influence of this variable hard to measure due to reverse causality and other sources of endogeneity [18]. The literature also shows that international accreditation, such as that from the Association to Advance Collegiate Schools of Business (AACSB), leads to substantial changes in the culture and processes of accredited business schools, making them more oriented towards research performance and the academic visibility of research [19–21]. More recently, the AACSB has also become more concerned by the impact of business education on society, which led to the development of new accreditation standards that require business schools to demonstrate societal impact [22]. While this relationship has been tested on its own, the independent effect of prestige while controlling for other spatial and country-level variables is less well-known. Thus we strongly hypothesise that:

**H1: Business school prestige–measured by university ranking and AACSB accreditation— is positively related to research performance (volume of publication and public visibility) after controlling for other organisational- and country-level factors.**

### University location as a predictor of research performance

In regional and urban science, scientific outputs—whether academic or industrial—are often conceptualised as part of a broader system of regional knowledge generation and innovation [23]. In this framework, all kinds of innovative activities (patents, inventions, research and development) are understood in a spatial context as the result of 'agglomeration benefits' from knowledge spillovers—i.e., new knowledge comes from people in proximity sharing ideas [24,25]. These benefits scale with city size, and thus are a primary driver of urban and economic growth [25–27]. The associated literature on regional 'systems of innovation', 'knowledge clusters', and 'innovative milieus' foregrounds the role of networks (including between universities and firms), relational trust, institutions, and regional cultures of competition and cooperation in generating new ideas and innovation [28–36].

A feature that has also received attention in the literature is the socio-spatial structure of research collaboration and citation networks. Researchers tend to collaborate and cite domestically [37,38] or with neighbours that share social and political characteristics [39–41] although this propensity is likely decreasing over time as globalisation increases and transportation/information costs decrease [41,42]. However, even in a globalised world, distance (spatial and temporal) continues to play an important role in enabling face-to-face communication [43], which is particularly important for knowledge-intensive sectors of economy, of which academia is a typical example.

These findings suggest the importance of fine-grained spatial proximities—among a range of proximities, including cognitive, social, organisational, and institutional—in structuring and explaining research output, as would be expected from the broader literature on agglomeration and knowledge spillovers [8]. However, to this point, relatively few studies analyse the importance of features at the organisational- or regional-scale. At the regional level,

agglomeration benefits to research activity have been observed, both in terms of 1) the role that spatial accessibility and distance play in predicting regional scientific activity [23] and collaboration frequency between regions [44,45], as well as 2) the impact of intraregional spatial concentration impact on knowledge production [46,47]. However, others have found that scientific output has largely decentralised since the 1980s (both within and across countries) with the rise of digital technologies and the ability to communicate online [48]. In this paper, we look at location in terms of spatial connectivity and hypothesise that:

**H2: Greater spatial connectivity of the business school location is positively related to research performance (volume of publication and public visibility).**

## Country-level predictors of research output

Research specifically on academic 'spatial scientometrics'—as distinct from understanding spatial patterns of knowledge creation more broadly—goes back to the 1970s, with an early focus on describing and explaining the distribution of publication and citation output by country [49]. Research output remains highly concentrated in OECD countries, and the US and UK in particular [4–9]. Several specific factors have been examined to explain this highly-skewed distribution. The overall development level and/or Gross Domestic Product (GDP) of a country is significantly associated with higher levels of research output and citations [5,6]. Similarly, national levels of research funding have demonstrated a high correlation with research activity. Pastor & Serrano [50] found that research and development (R&D) expenditure per capita is related to higher research output in EU countries, while Payne & Snow [51] show that, in the US, federal research funding increases publications, citations, and patents. Abbott and Doucouliagos [52] found a similar impact for research funding on importance-weighted publication output in Australia.

It is important to acknowledge that a large proportion of researchers are employed by higher education institutions, where their responsibilities are not limited by research, but also include teaching and administration. The lack of resources for any of these responsibilities can influence negatively on research performance–an effect recently seen during the COVID-19 pandemic, when research productivity of many academics suffered because of higher resource requirements for teaching in remote format. Therefore, it is helpful to look at the funding of the entire higher education system rather than narrowly defined research funding. Thus we hypothesise that:

**H3: Higher education funding in a country is positively related to the research performance (volume of publication and public visibility) of its universities.**

Beyond the effects of funding policies, the impact of immigration policies on scientific performance has also received attention in the literature. From a conceptual perspective, more open immigration policies both enable the talented researchers to relocate to a given country more easily (increasing the ability of institutions to attract better researchers), *and* provide for a more diverse population and urban milieu that is attractive for workers in the creative/knowledge economy. Empirically, the conversation in the literature has largely focused on the role of highly skilled migrants in STEM fields [53]. Some studies, however, looked more broadly at the contribution of foreign-born faculty to research performance of institutions employing them [54,55]. Based on those studies, we can tentatively hypothesise a positive relationship between recent immigration into the country and research performance.

**H4: Immigration into the country is positively related to the research performance (volume of publication and public visibility) of universities located in that country.**

Finally, recent literature on careers has started to look outside of the workplace to understand the additional predictors of employee motivation and performance. Research performance in particular is an outcome of work activities that are rarely happening strictly within the boundaries of a workplace, both spatially and temporally. The quality of life in the country might explain some additional variation in research outcomes because it influences how much of the cognitive bandwidth of researchers is occupied by everyday non-work-related stressors. For example, research in psychology showed that scarcity of a particular resource leads to excessive fixation on this resource, with less attention allocated to other matters [56]. The literature also highlights the importance of urban amenities, diversity, and tolerance for attracting creative knowledge workers [57,58]; therefore, we would expect the most talented researchers to want to live in countries with higher overall quality of life. Yet, we are missing empirical evidence connecting the quality of life in the country (measured in a more fine-grained way than GDP per capita) and scientific outcomes. In this paper, we operationalise quality of life through an index that combines average overall life satisfaction rating, average number of weekly hours worked, percentage of those in "very good or good" health (self-reported), and housing overcrowding rate. We tentatively assume that:

**H5: Higher quality of life in a country is positively related to the research performance (volume of publication and public visibility) of universities located in that country.**

Fig 1 summarizes our conceptual model.

In addition to the untested direct relationships which we have hypothesised, there remains a lack of clarity on the *relative* importance of meso- and macro-level features on predicting research outputs. To fill this gap, this paper conducts a multilevel analysis of variables at both the organisational- and country-level, including 1) business school prestige and spatial connectivity, and 2) a quality of life index, public spending on education, and recent immigration. At the same time, we recognise that researchers and institutions may respond differently to external features and incentives based on the development level of the country and its overall positioning within the global economic system. Thus we use a novel dataset of 497 institutions in

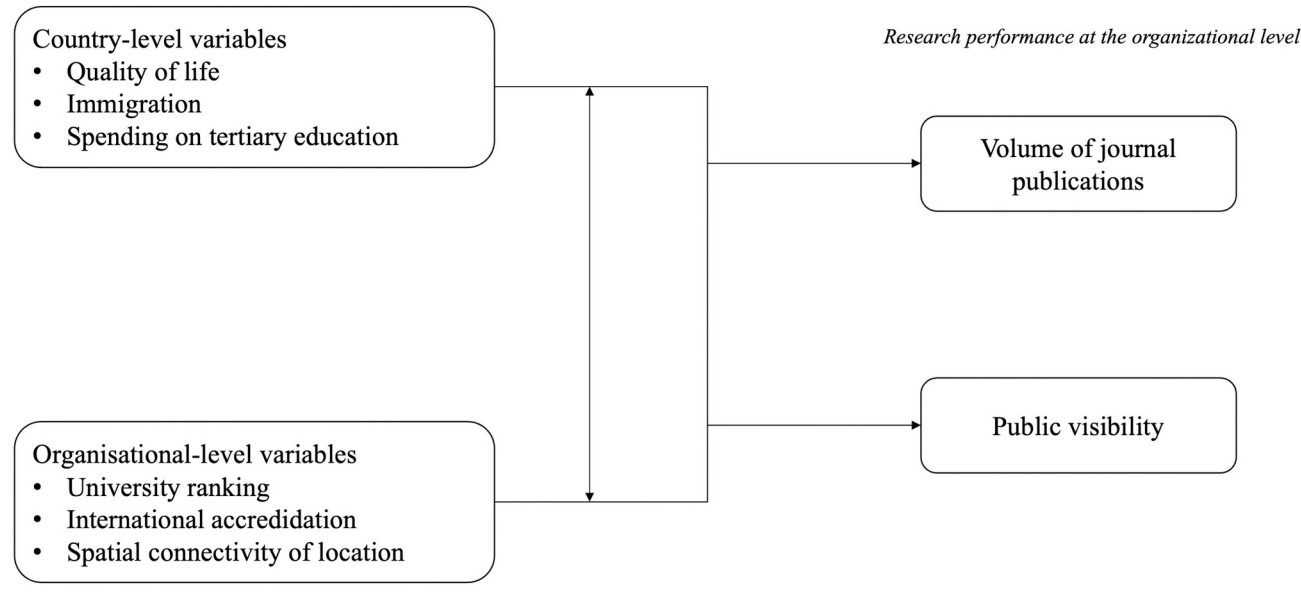

**Fig 1. Conceptual model of the association between country- and organisational-level factors and research performance.**

11 European countries, split between high- and low-income (those with GDP per capita above and below the EU average in 2018), to better understand the relationship between organisational- and country-level factors and the volume and visibility of business and management research. This leads to our final hypotheses that:

**H6a: The relationship between hypothesised features at the country-level (i.e., education spending, immigration, and quality of life) and the research performance (volume of publication and public visibility) of universities will vary based on characteristics of the organisations themselves (i.e., school prestige and spatial connectivity).**

**H6b: The relative importance of all hypothesised features (i.e., school prestige, spatial connectivity, education spending, immigration, and quality of life) on the research performance (volume of publication and public visibility) of universities will differ between high- and low-income countries.**

## Methodology

### Dataset

The data for this study was collected broadly following the approach adopted by Ryazanova et al. [59] in their study of research performance in the field of business and management. Top journals in the field of business and management were identified based on average 5 Year Impact Factors and citation coverage of the field. All institutions located in 11 European countries with publications in those journals over the period of 2007–2018 were included in the dataset (please see Table 1 below). The countries were chosen with the objective to obtain a sample diverse in multiple ways: it includes small (e.g., Ireland and Denmark), medium-sized (Norway and Sweden), and large (Spain and the UK) academic markets, as well as the mix of countries where English is the dominant language (Ireland and the UK) and those where it is not. It also includes countries with above- and below-average GDP per capita, to understand systematic differences in research performance by country-level development context. 2018 GDP per capita data come from Eurostat and are reported in Purchasing Power Standards (PPS). The values are indexed to 100, where 100 is equal to the EU average value.

### Dependent variables

In this paper, the dependent variables measured at the university level are research volume and public visibility of research. Data for research volume was sourced from the Web of Science.

**Table 1. Summary of the sample.** Countries Listed in Alphabetical Order.

| Country | No. institutions | Share institutions | 2018 GDP per capita | Income Level |
|---|---|---|---|---|
| Belgium | 26 | 5.2% | 118 | High |
| Denmark | 14 | 2.8% | 129 | High |
| Finland | 35 | 7.0% | 111 | High |
| Greece | 27 | 5.4% | 66 | Low |
| Ireland | 19 | 3.8% | 190 | High |
| Netherlands | 42 | 8.5% | 129 | High |
| Norway | 39 | 7.8% | 157 | High |
| Poland | 42 | 8.5% | 71 | Low |
| Spain | 77 | 15.5% | 91 | Low |
| Sweden | 32 | 6.4% | 120 | High |
| UK | 144 | 29.0% | 106 | High |
| **Total** | **497** | 100.0% | | |

*Research volume* is measured as the number of publications (articles and reviews only) published by 497 sampled organisations in 2016–2018 in academic journals. The selection heuristic for the journals follows the approach used by [59] in their global comparison of research performance in the field of Business and Management. These journals have the highest average 5 Year Impact Factor in 2007–2018, with their publications attracting 80% of all citations in this field (defined as journal listed by the Clarivate Journal Citation Reports under the categories "Business", "Business, Finance", "Management" and "Public Administration"). *Public visibility* is measured as a cumulative number of news mentions for the publications included in research volume variable (identified via DOIs) at the time of data collection (March 2022). This data was sourced from Altmetric Explorer (altmetric.com).

### Independent variables

To test hypothesis H1, we examine the impact of university ranking and business school accreditation on research performance. University *ranking* is measured through the Times Higher Education (THE) world university ranking for 2019. Rankings above 200 were categorised in ranges; for these universities, the midpoint of the range was used. For those ranked above 1000 (or 'unranked'), the value 1150 was used. Business school *accreditation* is identified through a dummy variable as to whether a given business school is accredited by the Association to Advance Collegiate Schools of Business (AACSB), the "longest-standing, most recognized form of specialized accreditation that an institution and its business programs can earn" [22]. Given the correlation between these variables, we have combined them into a single composite measure of *school prestige* by subtracting the z-score for THE rank (as it is reverse-ordered, i.e., lower numbers correspond to a higher rank) from the z-score for the AACSB accreditation dummy.

The second hypothesis (H2) describes, at an individual institution level, the agglomerative benefits that accrue to research activities based on spatial proximity. Given the collinearity between measures of spatial connectivity, we created a '*spatial connectivity*' index by combining the *z*-scores of three variables to proxy for agglomeration effects: the number of other universities (in the dataset) within 220km (essentially the minimum distance for every school to have at least 1 neighbour), distance to the nearest rail station (the 'RAILRDC' category from the 2022 'EuroGlobalMap' provided by EuroGeographics [60]), and distance to the nearest major airport ('Large' airports from the 'GISCO Airports 2013 dataset' provided by Eurostat).

The remaining hypotheses (H2-4) deal with the effect of the country-level societal context and funding on research performance. The choice of these specific variables is based on existing literature, as described earlier in the paper. The data on *public spending on tertiary education* (as a % of GDP) for each country in 2018 is obtained from the OECD (including both direct expenditures and public subsidies) [61]. In this paper, *quality of life* is measured through an index of combined *z*-scores for several variables collected by Eurostat [62]: average life satisfaction rating (0–10) for people 16 and over, the average number of weekly hours worked in primary job, the percentage of people 16 and over reporting self-perceived health as "very good or good," and the housing overcrowding rate for the entire population. For *immigration*, we use the percentage of recent immigrants aged 15–64 from any country foreign to the selected country from Eurostat [62]. Table 2 shows the relationship between each hypothesis and its corresponding independent variables of interest, data sources, and abbreviation. All variables are standardised (*z*-score) before being entered into the statistical model, which enables for easier comparison of effect sizes.

**Table 2. Variables, abbreviations, data sources, and associated hypotheses.**

| Level | Hypothesis | Name | Variable | Source | Years |
|---|---|---|---|---|---|
| 1 ($u$) | All (DV) | VOL | Research volume (# papers) | Web of Science | 2016–2018 |
| 1 ($u$) | All (DV) | NEWS | Public visibility (cumulative news mentions) | Altmetrics | 2016–2018 |
| 1 ($u$) | H1 | PREST | Business school prestige | | |
| | | | Times Higher Education (THE) overall ranking | THE | 2019 |
| | | | Business school accreditation status (dummy) | AACSB | 2018 |
| 1 ($u$) | H2 | CONNECT | Spatial connectivity index | | |
| | | | # of universities w/in 220km | Web of Science | 2018 |
| | | | Distance to nearest rail station | EuroGlobalMap | 2022 |
| | | | Distance to nearest large airport | Eurostat | 2013 |
| 2 ($c$) | H3 | EDSP | Public spending on tertiary education (% GDP) | OECD | 2018 |
| 2 ($c$) | H4 | IMM | % recent immigrants from any foreign country | Eurostat | 2018 |
| 2 ($c$) | H5 | QOL | Quality of life index | | |
| | | | Average life satisfaction rating (0–10) | Eurostat | 2018 |
| | | | Average weekly hours worked | Eurostat | 2018 |
| | | | % self-reporting "good" or better health | Eurostat | 2018 |
| | | | Housing overcrowding rate | Eurostat | 2018 |

## Statistical methods

Given this paper's interest in exploring the comparative and intersecting effects of variables at multiple geographic scales, a multilevel 'random effects' (RE) modelling approach is employed. When properly specified, RE models are particularly useful in this context: like fixed effects (FE) models, they control for the effect of grouping at multiple levels, but they also allow for the identification of the relationship between the dependent variable (at level 1) and specific variables at level 2, as well as cross-level interactions [63]. Fig 2 shows the modelling workflow used in this paper. The overall approach is exploratory in the sense that various model specifications and combinations of variables are tested to find the most parsimonious results, which is particularly important in multilevel models given the heightened effects of multicollinearity and the need to preserve variation across levels [64]. All models were estimated using the 'lme4' package in R. The detailed results of each modelling step, along with robustness checks, can be found in Appendix A.1-7 in S1 File.

First, we found the intraclass correlation coefficients (ICC) for both of our dependent variables of interest using RE 'null' models to gauge the overall variation caused by grouping the

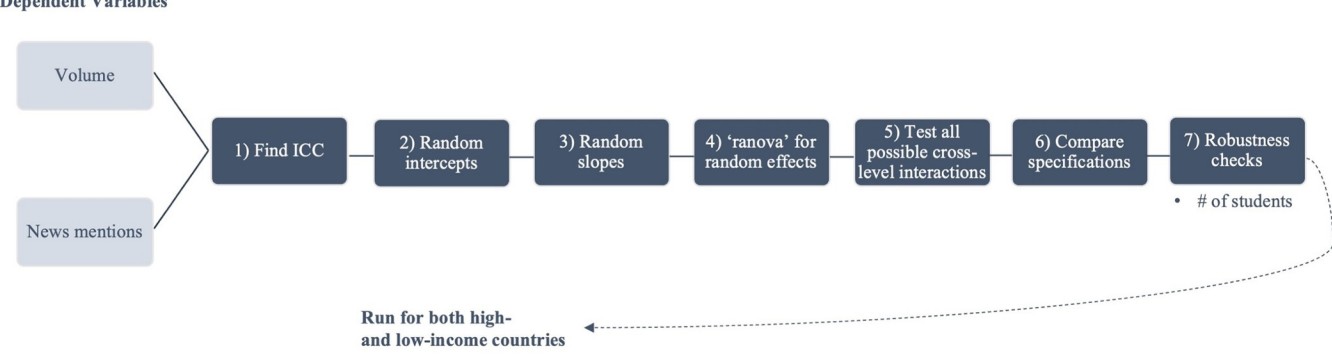

**Fig 2. Diagram of the modelling workflow.**

**Table 3. Intraclass correlation coefficients at country-level for each dependent variable and sample subset of interest.**

| Country-level ($c$) | VOL | NEWS |
|---|---|---|
| Full sample | 0.07 | 0.06 |
| High-income | 0.04 | 0.03 |
| Low-income | 0.21 | 0.03 |

data at the country level, according to the standard calculation in Eq (1):

$$\rho = \frac{\sigma_\alpha^2}{\sigma_\alpha^2 + \sigma_\varepsilon^2} \tag{1}$$

Here $\delta_\alpha^2$ is the variance attributed to the group-level random effect and $\delta_\varepsilon^2$ is the residual variance. Thus $\rho$ (the ICC) denotes the proportion of variance in the dependent variable attributed to a specific grouping scheme ($\alpha$).

As Table 3 shows, the overall ICC for research volume (0.07) and public visibility (0.06) indicates that a fairly substantial of the variance in these factors occurs at the country level. This is particularly true for research volume in the subset of low-income countries. Given this positive indication of the empirical importance of grouping at the country level, we decided to employ a two-level model (organisational- and country-level) in the analysis.

With a two-level approach selected, we then created a 'random intercepts' model for the both dependent variables ($y$) using the country ($c$)- and university ($u$)-level characteristics of interest. The specification, shown in Eq (2), used for estimating the random intercepts model used here is the 'random effects within-between' (REWB) [63]:

$$y_{uc} = \beta_0 + \beta_{1W}(x_{uc} - \bar{x}_c) + \beta_{2B}\bar{x}_c + \beta_3 z_c + (v_c + \varepsilon_{uc}) \tag{2}$$

where $\beta_{1W}$ specifies the 'within effect' on the group-demeaned (by country) value of $x$ and $\beta_{2B}$ specifies the 'between effect' on the grouped (by country) mean of $x$. In essence, the inclusion of both the group-demeaned and group-mean variables provides an estimate of the variation in a given level 1 predictor *within* level 2 groups *while controlling for* the difference in that predictor *between* groups. In fact, the estimate of the $\beta_{1W}$ coefficient in Eq (2) is exactly equivalent to the coefficient on the raw variable in a standard FE model. The difference is that the REWB allows for the inclusion of additional level 2 variables ($z_c$) whose variation is completely soaked up by the level 2 fixed effects in the standard FE model.

Another benefit of using the RE framework is that it is possible to model random *slopes* for each of the level 2 units to expose and better understand heterogeneity in the relationship between the $y$ and $x$ variables of interest. In this paper, each group-demeaned level 1 variable from Eq (3) is initially entered into a random slopes model, based on the general specification shown in Eq (3):

$$y_{uc} = \beta_0 + \beta_{1W}(x_{uc} - \bar{x}_c) + \beta_{2B}\bar{x}_c + \beta_3 z_c + v_{c0} + v_{c1}(x_{uc} - \bar{x}_c) + \varepsilon_{uc} \tag{3}$$

Fig 3 shows a diagrammatic example of the difference in model fit for research volume when accounting for (A) random intercepts when groups (countries, in this case) have structurally different values of a given $x$ variable (group-demeaned *PREST*, in this case), and (B) random slopes when the *direction* and/or *magnitude* of the relationship between $x$ and $y$ varies across groups. For simplicity, the visual examples in Fig 3 are built using only one independent variable of interest at a time, not the full final model specification used in this paper, so they cannot be directly interpreted. The overall magnitude and direction of effects generally follow

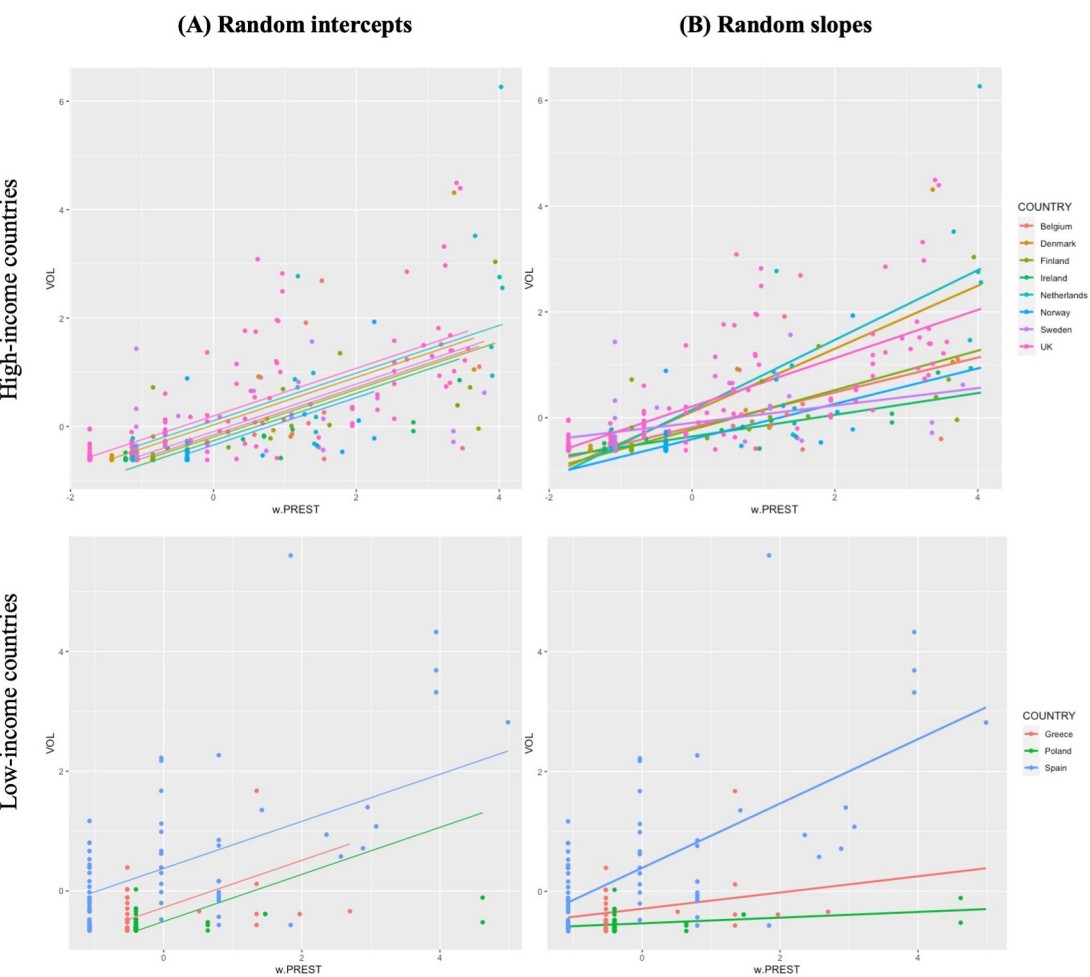

**Fig 3.** Visual example of the (A) Random Intercepts and (B) Random Slopes Relationships for Research Volume (VOL) and the Group-Demeaned Prestige (w.PREST) Independent Variable.

the final results, however: for example, universities in the Netherlands, UK, and Denmark tend to show steeper positive slopes for the *PREST* variable than in other countries, meaning that a decrease in prestige generally relates to a sharper dropoff in research volume there than in other countries. On the other hand, universities in Poland show a much flatter slope for the *PREST* variable, indicating that school prestige does not influence research volume to the same extent as in other countries.

Next, the 'ranova' function from the 'lmerTest' package in R is used to test the significance of each random effect term in the model to create the most parsimonious specification [65]. While the inclusion of random slopes provide for more precise estimates of the main effects on their own—and illuminate interesting heterogeneity in relationships across level 2 units (as Fig 3 demonstrates visually)—they also must be included in RE models that employ cross-level interactions to avoid bias in standard error estimates [66]. Given the significance of the random effects term provided by the 'ranova' function (in each case, for *PREST* only), each potential combination of cross-level interactions for both dependent variables were tested in turn, with the significant interactions retained (the full suite of interaction tests is shown in Appendix A.4-5 in S1 File). These specifications are shown in

Eqs (4)(A)–4(C):

$$
\begin{aligned}
y_{uc} = {} & \beta_0 + \beta_1 EDSP_c + \beta_2 QOL_c + \beta_3 IMM_c * (PREST_{uc} - P\bar{R}EST_c) + \beta_{4B} P\bar{R}EST_c \\
& + \beta_{5W}(CONNECT_{uc} - CON\bar{N}ECT_c) + \beta_{6B} CON\bar{N}ECT_c + v_{c0} + v_{c1}(PREST_{uc} \\
& - P\bar{R}EST_c) + \varepsilon_{uc}
\end{aligned} \tag{4(A)}
$$

$$
\begin{aligned}
y_{uc} = {} & \beta_0 + \beta_1 IMM_c + \beta_2 EDSP_c + \beta_3 QOL_c * (PREST_{uc} - P\bar{R}EST_c) + \beta_{4B} P\bar{R}EST_c \\
& + \beta_{5W}(CONNECT_{uc} - CON\bar{N}ECT_c) + \beta_{6B} CON\bar{N}ECT_c + v_{c0} + v_{c1}(PREST_{uc} \\
& - P\bar{R}EST_c) + \varepsilon_{uc}
\end{aligned} \tag{4(B)}
$$

$$
\begin{aligned}
y_{uc} = {} & \beta_0 + \beta_1 QOL_c + \beta_2 IMM_c + \beta_3 EDSP_c * (PREST_{uc} - P\bar{R}EST_c) + \beta_{4B} P\bar{R}EST_c \\
& + \beta_{5W}(CONNECT_{uc} - CON\bar{N}ECT_c) + \beta_{6B} CON\bar{N}ECT_c + v_{c0} + v_{c1}(PREST_{uc} \\
& - P\bar{R}EST_c) + \varepsilon_{uc}
\end{aligned} \tag{4(C)}
$$

This provides a stable point of comparison for the results between the two dependent variables, and gives insight into the way in which predictors of research volume and public visibility differ between high- and low-income countries. It is important to note that the fixed-effect model matrices for the low-income country models are rank deficient, so in each case the two 'between effect' coefficients and country-level *EDSP* coefficient are dropped automatically by the 'lmer' function. This may be due to the relatively low variability in values of *EDSP* across a small number of groups (3), or indicate that these models are saturated or suffering from multicollinearity.

Finally, robustness checks are made for each of the models with the number of students per school included to control for the effect of school size on research volume and public visibility. Due to parsimony and a lack of data availability (the number of students has been collected for only 332 out of the full sample of 497 schools), this is not used as the final specification, but the results of these robustness checks, shown in Appendix A.6-7 in S1 File, do generally confirm the results of the primary specification for the subset of schools in high-income countries. Again, for the low-income countries, inclusion of the number of students causes rank deficiency, and the coefficients for the between effects, *EDSP*, *IMM* are dropped from these models. The lack of observations with data on the number of students in this subset is likely the cause.

## Findings

### Spatial patterns of research volume and public visibility

To set the overall context for the analysis, Fig 4(A) and 4(B) shows the general spatial patterns for both variables of interest, along with boxplots of their distributions by country. For both research volume and public visibility, we can see that many of the high-performing schools are located in the high-income countries, and specifically in the UK, Belgium, and Netherlands. 15 schools show up in the top 25 for both research volume and visibility–of these, 8 are located in the UK, 3 in the Netherlands, 2 in Belgium, and 1 each in Finland and Denmark. Erasmus University in the Netherlands is ranked highest in the dataset for both variables; other well-known universities, such as London School of Economics, Oxford University, Cambridge University, Katholieke University Leuven, and the University of Amsterdam also appear in the top 25 for both variables.

Overall, the spatial pattern for research volume appears to be somewhat more evenly-distributed than for public visibility; while the high-performing schools for both variables are relatively tightly spatially-clustered within the UK-Belgium-Netherlands triangle, this clustering is even more pronounced for public visibility. This may be at least partially due to the relatively

## (A) Spatial Patterns

### (i) Research Volume

### (ii) Public Visibility

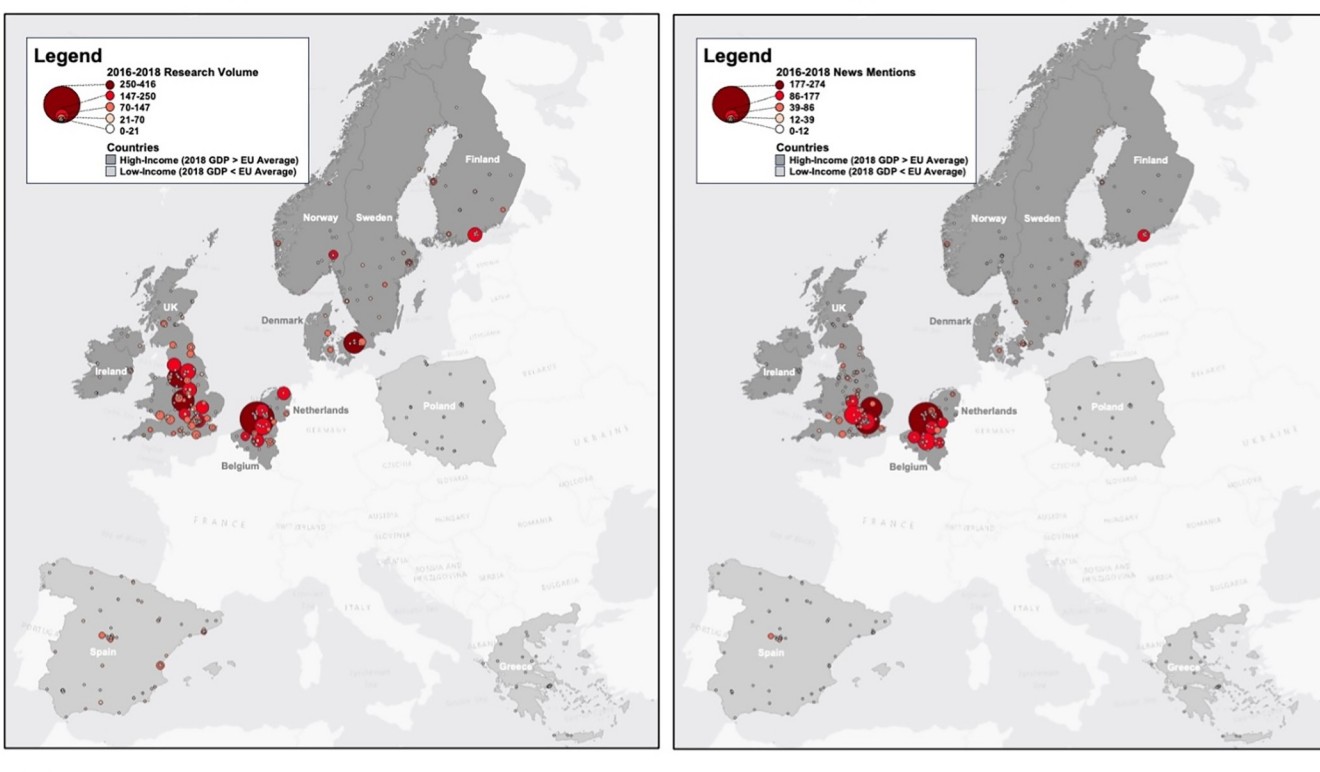

## (B) Boxplots

### (i) Research Volume

### (ii) Public Visibility

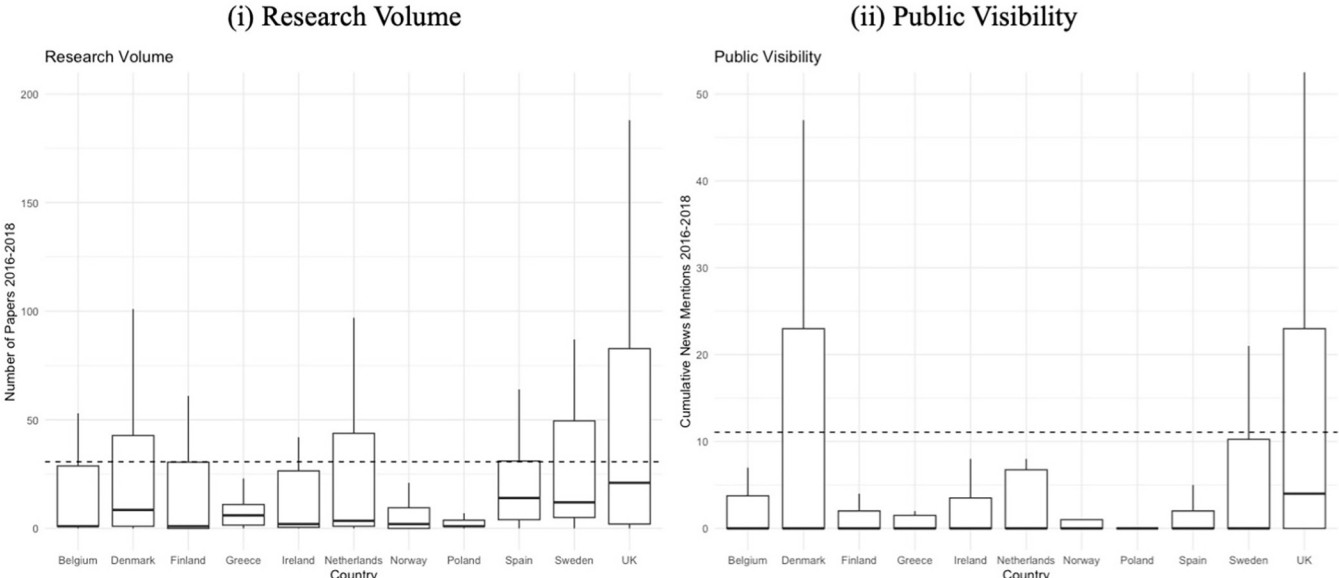

**Fig 4.** Map Showing (A) Spatial Patterns and (B) Boxplots of the Distribution of (i) Research Volume and (ii) Public Visibility by University. On (B), the Overall Mean Value is Marked in a Dashed Line.

large media markets in this area (i.e., London and Amsterdam), or could be a product of the higher concentration of high-prestige universities here. The variance of public visibility within each country is also larger than for research volume.

While the UK has the largest distribution of values (and highest upper quartile values) for both variables, a few high outliers appear to be pulling up the distribution of public visibility by school in Denmark, and, to a lesser extent, Sweden, when compared to research volume.

## Model results

The results described here focus on the outputs of the *final specifications* laid out in Eqs (4)(A) and (4)(B). A note on terminology before we begin: the use of verbs here such as 'benefit,' 'affect,' 'impact,' etc., is not meant to imply a causal statistical relationship, as the methods used in this paper cannot identify if causality exists or not. These terms are chosen because using them eases the explanation of complex interaction effects for the reader, which are much harder to describe clearly if they are constantly caveated by non-causal terminology. None of the results in this paper are causal in nature. Fig 5(A) and 5(B) shows the regression coefficients, 95% confidence intervals, significance of the independent variables, and directions of effect for tested interaction terms for the *research volume* dependent variable split between the high- and low-income subsets of countries; Fig 6(A) and 6(B) shows the same for the *public visibility* dependent variable. The specifications shown in these Figs include two cross-level interactions with *PREST*: for *IMM* and *QOL*. The other specification–Eq (4)(C)–which included an interaction between *PREST* and *EDSP*, was not significant in any context, so was not reported in the main section of the paper. However, these results (and the full set of intermediary results) can be found in Appendix A.1-5 in S1 File.

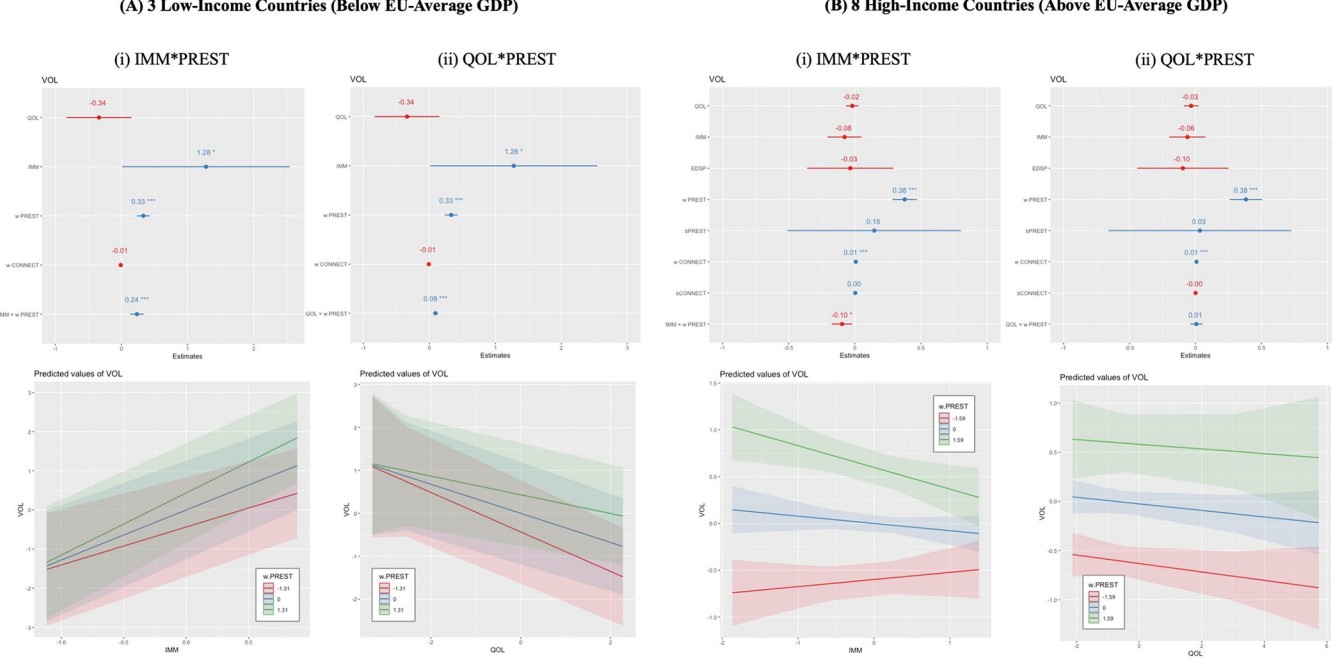

**Fig 5.** Regression Coefficients, 95% Confidence Intervals, Significance of the Independent Variables, and Tested Interaction Effects for the Research Volume Dependent Variable for (A) Low-Income and (B) High-Income Countries. Results are shown for the (i) Percent Recent Immigrants (IMM) x School Prestige (PREST) and (ii) Quality of Life Index (QOL) x PREST Specifications.

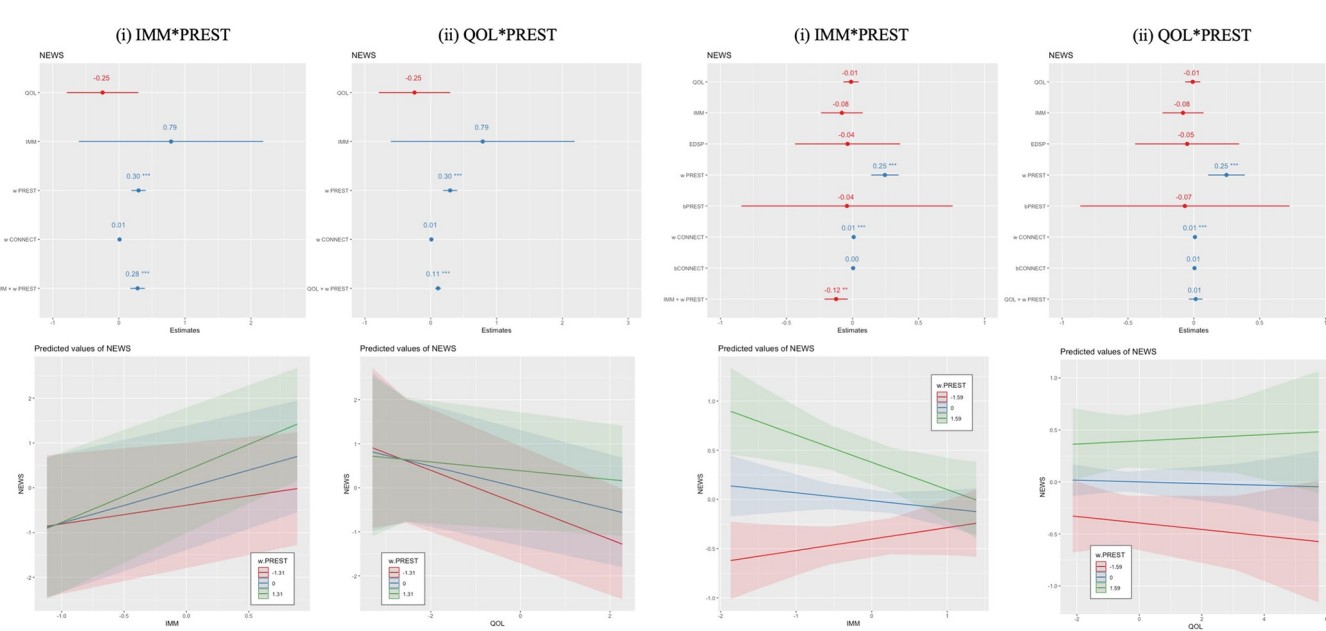

**Fig 6.** Regression Coefficients, 95% Confidence Intervals, Significance of the Independent Variables, and Tested Interaction Effects for the Public Visibility Dependent Variable for (A) Low-Income and (B) High-Income Countries. Results are shown for the (i) Percent Recent Immigrants (IMM) x School Prestige (PREST) and (ii) Quality of Life Index (QOL) x PREST Specifications.

Table 4 connects these results to each of the paper's specific hypotheses. First, as expected based on the existing literature, business school prestige is significantly related to research volume and public visibility (H1). Second, higher levels of spatial connectivity–in terms of proximity to other universities in the dataset, air, and rail services–relate to significantly higher values of research volume and public visibility in high-income countries. This confirms the importance of agglomeration (broadly conceived) to research activity and dissemination in these countries, and lends partial support to both H2 and H6b (given the differentiation between low- and high-income countries).

**Table 4. Hypotheses, associated variables, and observed direction and significance of the relationship.**

| | | | Results (Low-/High-Income Countries) | |
|---|---|---|---|---|
| **Hypothesis** | **Variable** | **Direction** | *Volume* | *Public Visibility* |
| H1 | PREST | + | +/+ | +/+ |
| H2 | CONNECT | + | (ns)/+ | (ns)/+ |
| H3 | EDSP | + | (ns)/(ns) | (ns)/(ns) |
| H4 | IMM | + | +/(ns) | (ns)/(ns) |
| H5 | QOL | + | (ns)/(ns) | (ns)/(ns) |
| H6a | EDSP*PREST | | (ns)/(ns) | (ns)/(ns) |
| | IMM*PREST | | +/- | +/- |
| | QOL*PREST | | +/(ns) | +/(ns) |
| | EDSP*CONNECT | | (ns)/(ns) | (ns)/(ns) |
| | IMM*CONNECT | | (ns)/(ns) | (ns)/(ns) |
| | QOL*CONNECT | | (ns)/(ns) | (ns)/(ns) |

The main effects of the country-level variables, on the other hand, display some unexpected relationships with our dependent variables of interest. Public spending on tertiary education does not show a statistically significant relationship to either of the dependent variables, which does not support H3. Similarly, the main effect of recent immigration is only significantly related research volume in low-income countries; hence H4 is not strongly supported either. In addition, the main effect of the quality of life index (QOL) is likewise not significant for either of the dependent variables or subsets of countries, negating H5.

However, these results come into clearer focus when we examine their interaction with school prestige (partially confirming H6a, although no specific direction of relationship was hypothesised). The key set of significant results for the interaction effects are between 1) recent immigration and business school prestige for both dependent variables and subsets of countries, and 2) quality of life and prestige for both dependent variables in low-income countries only. The direction of these interaction effects is shown in Figs 5 and 6: in low-income countries, more prestigious business schools benefit to a greater extent from higher levels of recent immigration–in terms of better research performance (for both volume and visibility)–than less prestigious schools, although the direction of relationship is positive across the board.

Interestingly, in high-income countries, the opposite relationship exists: *less prestigious* schools benefit the most from higher levels of recent immigration. In addition–counterintuitively—in low income countries the quality of life index is *negatively* related to research performance, although more prestigious schools are less negatively impacted (creating a positive effect for the interaction term). The relationship is not significant in high-income countries. Finally, none of the cross-level interactions involving spatial connectivity are significant due to the lack of significance of that variable as a random effect, as discussed above.

## Discussion and conclusion

Our work shows the multi-level nature of factors leading to spatial inequality in the distribution of research outputs, and the important role of both organisational- and country-level predictors in determining research performance. In this study, country-level factors explain 7% of the overall variance in research volume and 6% of the variance in public visibility between business schools. In addition, the direction and significance of various predictors of research performance differ between the subset of schools in 'low-income' (below EU-average GDP per capita) vs. 'high-income' (above EU-average GDP per capita) countries. This supports the idea (generally) that research performance is driven at least in part by national contextual factors, and could be the result of several over-arching mechanisms. First, it might be impacted by research culture in the country, specifically, the extent to which it is oriented towards top-ranked journals [67]. Secondly, the historical location of top journals in some countries and not in others might result in a better access to these journals from within this country. Thirdly, the country-level variation might capture the language effect (whether the country is English-speaking or not), which is particularly important for publications in top journals in business and management. These journals require the level of theoretical development and compelling narratives which are much harder to achieve for authors who are non-native English speakers [68]. Importantly, this finding points to the value of understanding the role that country-level context plays in fostering research performance overall, which is often ignored in individual-level analyses.

Interestingly, our results show that the effect of country-level variables on research outcomes occurs primarily through their interaction with business school prestige (measured by business school accreditation and university ranking), rather than directly, and that there are significant differences in the direction of these relationships between low- and high-income

countries. At a broad level, this means that the influence of contextual factors on researchers varies considerably based on the type of institution and the development level of the country they are located in. In low-income countries, *more prestigious* schools benefit the most from higher rates of recent international immigration from any foreign country (although all universities benefit to some extent). This could indicate that more prestigious schools in low-income countries are best-known and best-positioned to benefit from increased rates of recent immigration from any foreign country, and/or that these schools tends to be the kind of internationally-oriented institutions (e.g., with teaching options in English) that immigrants might consider working at.

In these countries a composite measure of a country's quality of life (including self-rated life satisfaction, health, working hours, and housing overcrowding) is also positively related to research outcomes through its interaction with school prestige (see Figs 5(A)(ii) and 6(A)(ii)). This means that in low-income countries with *lower* quality of life, there is not much difference in the production of researchers at business schools with different levels of prestige; but in low-income countries with *higher* quality of life, researchers at less prestigious business schools produce significantly less than those at more prestigious business schools. This was an unexpected finding from a theoretical standpoint. However, there could be a general correspondence between these largely self-identified ratings of (country-level) quality of life and workplace-cultural features at less prestigious institutions that are in some ways more market-oriented, less communal, and more competitive, which result in the observed higher research performance coupled with lower quality of life. These workplaces offer an extrinsic motivation for aiming towards the highest standard of performance–especially given the much wider disparity between the top and bottom of the professional scale in these countries–but at the cost of lower quality of life. Such trade-offs point towards short-term orientation in academic careers, which then might lead to negative career outcomes in the longer term [69]. Given that the negative effect of quality of life on research performance is not present to the same extent for more prestigious schools, this may also reflect the fact that academics at more prominent universities in these countries can trade off institutional prestige for the productivity benefits of a hyper-competitive environment. In any case, this counterintuitive finding deserves more attention in future research on this topic.

Interestingly, at the organisational-level, we found that greater spatial connectivity–operationalised through proximity to nearby business schools, rail stations, and airports–is not significantly linked to higher levels of research volume and public visibility for research in low-income countries. In some ways this is simply a reflection of the geography of business school locations in this sample of countries; e.g., if the most prestigious schools with the highest research performance are located in the capital city, but that city is relatively far away from the country's other major cities and universities, our spatial connectivity variable will not achieve significance. At the same time, this is a useful result, because it indicates that these agglomeration effects are *not* a primary mechanism driving research performance in these countries; rather the most prominent, prestigious, and (perhaps) internationally-oriented universities (wherever they are located) are driving research outcomes. From a policy perspective, this is also a reflection of the fact that these countries generally only have one or two prominent, research-oriented business schools; if the overall density of schools were increased, even within a particular urban area, perhaps the agglomeration effects observed in high-income countries would begin to take effect.

In high-income countries, spatial connectivity *is* a significant predictor of research performance. This supports much of the existing research on agglomeration in knowledge production generally–and scientific research more specifically–which has found that spatial accessibility increases knowledge exchange and collaboration frequency [23,37–40,44,45]. As

mentioned above, it may be that a 'critical mass' of well-connected high-performing organisations, as is the case in high-income countries, is necessary for agglomeration effects to take root. In fact, it is possible that the agglomeration effects resulting from high concentrations of population, knowledge-creating organisations, and transportation infrastructure is what made these countries high-income in the first place. While more detailed research on these effects is needed, by studying these relationships at the individual institution (rather than regional) scale, and assessing differences across countries at different stages of development, this finding adds to our understanding of the role of local agglomeration factors on research performance.

At the country-level, the effect of recent immigration on research performance in these countries also occurs primarily through its interaction with business school prestige. However, unlike in the low-income subset, in high-income countries *less prestigious* business schools appear to benefit the most from higher rates of recent international immigration. This could be due to the fact that less-established business schools must rely more heavily on the national urban-economic milieu—and general factors that benefit all businesses in attracting highly skilled workers, such as Florida's [57] three 'Ts' of technology, talent, and tolerance—when recruiting new staff, while more established schools have enough international recognition that they can recruit competitively despite, e.g., less welcoming environments for immigrants. In other words, being located in a country which is more welcoming for immigrants has a "halo effect" on the schools that do not have a strong "employer brand" [70] and helps them to attract better human capital and grow their research volume as well as public visibility of their research. This effect could also be driven by universities located in countries with smaller domestic education markets, because those might experience the deficit of high quality entrants into research careers due to limited local supply of PhD graduates.

It is important to note that this analysis is not without limitations. Due to the constraints inherent in the data, we were only able to compare business schools across 11 European countries, which–while selected to provide a range of academic, infrastructural, and cultural conditions–inherently limits the generalisability of the results. Future work involving data from other scientific disciplines and a wider range of low- and high-income country contexts–while difficult to obtain–would be particularly valuable for assessing research questions related to the role of cultural features and government policy on research performance. Additional control variables, including an expanded range of quality of life indicators, proximity to a wider range of research centres and/or research-oriented enterprises, and the number of faculty at each business school, while not readily available for this analysis, would also be useful to include in future extensions of this work. Similarly, future data collection on the characteristics of *individual researchers* could be combined with the characteristics of the business schools and countries collected here to provide a more holistic understanding of the interplay between the individual, organisational, and contextual features driving research performance. Methodologically, the importance of parsimony in specifying the multilevel models also limits the number of variables that can be included (and thus relationships that can be tested)–and also required us to combine several variables into aggregated indices, which slightly obscures the direct mechanisms driving the observed relationships. A particularly interesting future research avenue for identifying causal mechanisms could involve a randomised controlled trial (RCT) or quasi-experimental research design in which research outcomes for universities that have been 'treated' by a given policy or cultural context are compared to 'controls' that are similar in as many ways as possible but lack the hypothesised exposure, e.g., changing immigration policy or increased public education spending.

Despite these limitations–and keeping in mind that the results pertain specifically to business and management researchers in 11 European countries using the available data collected–we think that this analysis can have several important implications for local and national

policy. First, the results show that investment in regional infrastructure and connectivity, such as rail and air connections, has a benefit to scientific output and its public dissemination. While this result is significant only for high-income countries, this is likely because the low-income countries studied here lack a 'critical mass' of well-connected universities (and other knowledge-creating organisations) to generate observable agglomeration effects. At a national level, the results also indicate that more open immigration policies benefit scientific output for less-prestigious business schools in high-income countries, providing a direct policy pathway to increasing research output for universities that aren't already well-known enough to attract the most talented researchers. In low-income countries, recent immigration rates are even stronger predictors of research performance across all levels of institutional prestige, and thus more open immigration policies would likely benefit research performance in these countries to an even greater extent.

In addition, the paper's results suggest that, in low-income countries, the lower the quality of life, the more researchers are incentivised to produce higher levels of research output. While this may in part reflect the greater disparities inherent in these countries' economic systems, it is noteworthy–and perhaps concerning–that we have observed a negative correlation between country-level quality of life and research performance in these countries, which is particularly felt by researchers at less prestigious institutions. If researchers at more prestigious schools in these counties are, in fact, able to mitigate this effect (as our results suggest) by trading prestige for hyper-competitiveness, it would be worthwhile to further examine the mechanisms driving the prestige premium and attempt to provide supports that (partially) substitute for those benefits to researchers at lower prestige universities. This, in theory, would allow them to somewhat mitigate the negative effects of quality of life on research performance. In general, while this connection certainly deserves further research, the preliminary evidence presented here raises an important normative and political question: if researchers produce more articles in lower quality-of-life environments, is the trade-off ultimately worth it?

## Supporting information

**S1 File. This file contains Appendix A.1-7.**
(DOCX)

## Author Contributions

**Conceptualization:** Kevin Credit, Olga Ryazanova, Peter McNamara.

**Data curation:** Kevin Credit, Olga Ryazanova, Peter McNamara.

**Formal analysis:** Kevin Credit.

**Investigation:** Kevin Credit, Olga Ryazanova.

**Methodology:** Kevin Credit.

**Visualization:** Kevin Credit.

**Writing – original draft:** Kevin Credit, Olga Ryazanova.

**Writing – review & editing:** Kevin Credit, Olga Ryazanova, Peter McNamara.

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
