## [Decision Letter · Decision Letter 0]

30 Nov 2023

PONE-D-23-26138Spatial Connectivity, Public Investment, and Immigration: The role of university- and country-level factors in the volume and visibility of business and management researchPLOS ONE

Dear Dr. Credit,

Thank you for submitting your manuscript to PLOS ONE. After careful consideration, we feel that it has merit but does not fully meet PLOS ONE’s publication criteria as it currently stands. Therefore, we invite you to submit a revised version of the manuscript that addresses the points raised during the review process. This is a timely and fine research piece, but it needs a major revision from several angles. Both reviewers provided a very detailed critical feedback with important comments that need to be addressed properly. As Rev2 points out, both the Introduction and the methodological parts require refinement. Also, for the broader geopolitical picture (esp. connected with HE), some fresh pieces of scientific papers should be looked at, referred to critically and embedded in the body of the analysis. See these, for instance: https://www.tandfonline.com/doi/abs/10.1080/14767724.2023.2166465https://link.springer.com/book/10.1007/978-3-030-94415-5https://ecpr.eu/Events/Event/PanelDetails/9963https://www.tandfonline.com/doi/abs/10.1080/03098265.2022.2045572

We look forward to receiving your revised manuscript.

Kind regards,

István Tarrósy, PhD

Academic Editor

PLOS ONE

Journal Requirements:

3. Please ensure that you include a title page within your main document. You should list all authors and all affiliations as per our author instructions and clearly indicate the corresponding author.

4. Please amend your manuscript to include your abstract after the title page.

5. We note that Figure 4 in your submission contain mapsatellite] image which may be copyrighted. All PLOS content is published under the Creative Commons Attribution License (CC BY 4.0), which means that the manuscript, images, and Supporting Information files will be freely available online, and any third party is permitted to access, download, copy, distribute, and use these materials in any way, even commercially, with proper attribution. For these reasons, we cannot publish previously copyrighted maps or satellite images created using proprietary data, such as Google software (Google Maps, Street View, and Earth). For more information, see our copyright guidelines: http://journals.plos.org/plosone/s/licenses-and-copyright.

a. You may seek permission from the original copyright holder of Figure 4 to publish the content specifically under the CC BY 4.0 license.  

7. Please note that PLOS ONE has specific guidelines on code sharing for submissions in which author-generated code underpins the findings in the manuscript. In these cases, all author-generated code must be made available without restrictions upon publication of the work. Please review our guidelines at https://journals.plos.org/plosone/s/materials-and-software-sharing#loc-sharing-code and ensure that your code is shared in a way that follows best practice and facilitates reproducibility and reuse.

Reviewers' comments:

Reviewer's Responses to Questions

**Comments to the Author**

1. Is the manuscript technically sound, and do the data support the conclusions?

Reviewer #1: Partly

Reviewer #2: Partly

2. Has the statistical analysis been performed appropriately and rigorously? 

Reviewer #1: Yes

Reviewer #2: Yes

3. Have the authors made all data underlying the findings in their manuscript fully available?

Reviewer #1: Yes

Reviewer #2: Yes

4. Is the manuscript presented in an intelligible fashion and written in standard English?

Reviewer #1: Yes

Reviewer #2: Yes

5. Review Comments to the Author

Reviewer #1: In general, it can be ascertained that the study deals with a very relevant topic, at an appropriate professional level, primarily based on quantitative research methods. All in all, and taking into account the comments below, the study is suitable for publication with minor revisions and modifications.

The introduction outlines the scope of the study and also points out its novelty and some of its practical and theoretical aspects, however, this part gives the impression of a more detailed and extensive abstract, or a summary of the research results (than an introduction), as it describes the aims of the study, addresses its main methodological issues, and presents the conclusions (at the same time, it contains fewer literature antecedents and references), which isn’t an issue if the editorial board accepts this approach.

In connection with this chapter, the reviewer recommends that the authors explain in more detail who has dealt with this topic in the past, with what methods, and what the most important findings of previous research related to the topic. It would be appropriate to close the chapter by presenting the objectives of the research.

The study partially fulfils the above „task” in the chapter titled "Spatial distribution of research performance", which essentially details the theoretical framework of the research topic, as well as clarifying the objectives and presenting and formulating the hypotheses. At the same time, the range of sources could be expanded here, especially in connection with findings related to innovation (for example Asheim, B. T., Coenen, L. (2006): Contextualising regional innovation systems in a globalising learning economy: On knowledge bases and institutional frameworks. The Journal of Technology Transfer, 1., 163-173. https://doi.org/10.1007/s10961-005-5028-0; Bathelt, H., Malmberg, A., Maskell, P. (2004): Clusters and knowledge: Local buzz, global pipelines and the process of knowledge creation. Progress in Human Geography, 1., 31-56. https://doi.org/10.1191/0309132504ph469oa; Godin, B., Lane, J. P. (2013): Pushes and pulls: Hi(S)tory of the demand pull model of innovation. Science, Technology & Human Values, 5., 621-654. https://doi.org/10.1177/0162243912473163; Oláh, D. ; Alpek, B. L. (2021): The theoretical model of spatial production for innovation. Journal of Innovation and Entrepreneurship 10 (1) https://doi.org/10.1186/s13731-021-00182-4).

The finding regarding the relationship between quality of life and academic performance is one of the surprising results of the research. In this context, it would be worthwhile to present the authors' approach to the quality of life when describing the hypotheses. Later, the study provides an answer to this in the methodological part, but since many factors not included in the present study, but related to the quality of life, are also mentioned in the description of the hypothesis, it would be worthwhile to emphasize the approach used in the study.

The presentation of the methods is precise and detailed, and the methodological approach is appropriate. However, the section can be further developed along the lines of the following findings. On the one hand, the steps and methodological solutions included in the source cited by the authors should be presented in more detail. In connection with the selection of the sample, it should be made clearer why these countries were chosen and why the authors decided not to investigate (at least) the EU countries, either by expanding the range of journals or countries. Why is it missing from the analysis, for example, France or Germany? It is conspicuous to the reader that the weight of the United Kingdom is strikingly high in terms of the number of institutions among the selected countries (40%), and that all the selected countries are very close to each other in terms of development and quality of life (nearly the same HDI, etc.). The question arises whether, for example in terms of the quality of life: To what extent would the results of the models differ from the current ones if countries with a lower quality of life were also included in the sample?

The description of the indicators and calculation procedures is precise and detailed. It is particularly useful from the point of view of the results that the authors describe the range of selected indicators and sub-indicators in a clear form, and they are also reviewed in a summary table. Either as a future research direction or in the context of this research, it would be worthwhile to also take into account the proximity and number of other institutions directly or indirectly related to research performance (for example research centres, innovation centres, the size of the number of enterprises operating in the high-tech sector), and expand the range of indicators in connection with the measurement of quality of life. Regarding the robustness check, the authors mention that “Finally, robustness checks are made for each of the final models with the number of students per school…”. How much it would have been more suitable, and how much it would have led to other results using the number of lecturers/researchers instead of the number of students?

The presentation of the results is accurate and clear. A special advantage is that the study emphasizes the limitations of the research in detail and clearly, thereby allowing the reader to interpret the study's results in an appropriate context. In addition to all of this, it would be worthwhile to draw attention to the content of the indicators even more prominently in the introductory and summary sections (as the authors do with the hypotheses). In addition, it could be emphasized even more that the investigation focuses on a special scientific field and region.

In my opinion, it could be an interesting and useful research direction for the extension of the research to other scientific fields and regions.

About figures and tables, it would be recommended to insert a summary line in Table 1, as well as display the share of each country within the sample. The texts of the figures are very small and therefore difficult to read properly. Similarly, the maps and texts in Figure 4 are difficult to see. All in all, the tables and figures in the study are informative and easy to interpret.

In general, the study deals with an interesting and useful topic, based on a suitable methodological base. Although it would be interesting and worthwhile to expand the horizon of the research in several aspects with minor modifications, the study is suitable for publication even in its current form.

Reviewer #2: Spatial Connectivity, Public Investment, and Immigration: The role of university- and country-level factors in the volume and visibility of business and management research

This paper focuses on the examination of the role of universities, research centres, organizations that play role in helping scholars to achieve better results. Educational and research institutions can be examined from different point of views like spatial constellation, their role at different levels, moreover being an engine of regional and local collaborations, innovations and initiatives, an outstanding potential employer, probably the largest one. To me authors undertake too much factors in the introduction. As the title also indicates, too long, too complicated, readers including me do not understand the point of their paper clearly. My suggestion would be for the authors to clarify the purpose and focus of the study by attempting to define a narrower scope.

In the first sentence on the 3rd page authors refer to the literature that explores a broad range of organizational factors, but without citation. They should put the most relevant citations/references over there, what sort of literature analyze the aforementioned factors actually. Later on they could refer back to these review even picking the current citation or statement they have mentioned.

In regard to H1 assumption I am afraid I need to say this is an axiom, this is not a real question/hypothesis.

In the case of H2, I think this assumption must be better, because positive implications of funding, subsidies, financial support could already be measured. However there are more than 25,000 universities across the world. There is no one research centre that could develop models for analysing all of them. Conditions, circumstances, endowments are so deeply different, even if you dealt with the American, or British, or both system, but universities in general? If you might fix your intro and hypothesis, make them much more precise - by adding clarifying thoughts - your statements would be also more valid.

About H3. Authors use the concept "immigration", but they should clarify again that migration of who? What kind of people, from which areas? Scientists? Or physical labor force? Or students? Or qualified labor force like doctors, engineers, researchers?

In Methodology authors already refer to the countries they have picked for having a picture on universities' role, but these countries are European countries, more precisely the Scandinavian countries, the British islands and Spain as a cuckoo egg. I am afraid I do not understand why authors compare these state's universities, why they have not chosen the Netherlands or Belgium, or even Germany as a subject for their research.

If authors clarify all problematic parts (introduction and hypotheses), their paper's quality will be much higher, much greater therefore I recommend them to fix these shortcomings and defaults.

6. PLOS authors have the option to publish the peer review history of their article (what does this mean?). If published, this will include your full peer review and any attached files.

Reviewer #1: No

Reviewer #2: No

---

## [Author Response · Author response to Decision Letter 0]

15 Feb 2024

We would like to thank the editorial team and the referees for your detailed, helpful and constructive feedback on our paper “Spatial Connectivity, Public Investment, and Immigration: The role of university- and country-level factors in the volume and visibility of business and management research.” We greatly appreciate the time and effort you invested in evaluating our work, and your comments have been immensely helpful in improving the quality of our research.

We have carefully considered each of your comments, discussed them thoroughly, and incorporated them into our latest draft. As you will see in the revised version of the paper, the response to these comments - in particular, the relatively small sample of countries previously examined that both reviewers mentioned - has fundamentally shifted our perspective on the analysis and changed the paper substantially. In the revised manuscript we have included business schools from 4 additional countries in our analysis, which allows us to explicitly leverage the differences in national development level (by splitting the sample into ‘low’- and ‘high’-income countries) to better contextualise and understand the results. This expansion of the analysis (i.e., every result now has a low-income and a high-income version) necessitated some other changes: two new hypotheses have been added to recognise the fact that institutions may respond differently to external factors based on the development level of the country and its overall positioning within the global economic system and highlight the role of prestige in our conceptual thinking. To conserve space and provide a more coherent narrative (in light of the expanded scope of the paper), we have also now removed the analysis of the academic visibility variable. 

Interestingly, the new results for ‘high-income’ countries are basically in line with our original results. But now, by splitting the sample - and including two additional low-income countries - we can see that there are substantial differences in the results for ‘low-income’ countries. Overall, we think that this new approach significantly improves the paper and enhances what we can take away from the analysis.

In addition, we have contextualised the paper in terms of the new geopolitics of higher education, further clarified some of the indicators used in the analysis, made changes to the figures and tables, and edited the title and abstract to reflect the results of the new analysis.

We attach the most recent draft of our paper (with changes tracked) and a detailed response to each of the editor's and reviewers' specific comments and concerns to demonstrate how we have incorporated their feedback.

Once again, thank you for your valuable contributions to our research.

---

## [Decision Letter · Decision Letter 1]

27 May 2024

The role of organisational- and country-level factors in the volume and public visibility of business and management research

PONE-D-23-26138R1

Dear Dr. Credit,

We’re pleased to inform you that your manuscript has been judged scientifically suitable for publication and will be formally accepted for publication once it meets all outstanding technical requirements.

Kind regards,

István Tarrósy, PhD

Academic Editor

PLOS ONE

Additional Editor Comments (optional):

Both reviewers acknowledged that the original manuscript has been improved substantially and their comments and questions have been dealt with profoundly. I can support the final decision to accept the paper for publication.

Reviewers' comments:

Reviewer's Responses to Questions

**Comments to the Author**

1. If the authors have adequately addressed your comments raised in a previous round of review and you feel that this manuscript is now acceptable for publication, you may indicate that here to bypass the “Comments to the Author” section, enter your conflict of interest statement in the “Confidential to Editor” section, and submit your "Accept" recommendation.

Reviewer #1: All comments have been addressed

Reviewer #2: All comments have been addressed

2. Is the manuscript technically sound, and do the data support the conclusions?

Reviewer #1: Yes

Reviewer #2: Yes

3. Has the statistical analysis been performed appropriately and rigorously? 

Reviewer #1: Yes

Reviewer #2: (No Response)

4. Have the authors made all data underlying the findings in their manuscript fully available?

Reviewer #1: Yes

Reviewer #2: Yes

5. Is the manuscript presented in an intelligible fashion and written in standard English?

Reviewer #1: Yes

Reviewer #2: Yes

6. Review Comments to the Author

Reviewer #1: First of all, I would like to thank the authors for their work on the study and their responses to the peer review. In my opinion, the study in its current form is generally suitable for publication. I would like to make only two minor suggestions regarding the latest version.

Firstly it should be explained at the very beginning of the study, in the abstract, what the study considers a high- and low-income country. This will be done later, but in many cases, the reader will focus first (or exclusively) on the abstract and may misunderstand the meaning of the concepts due to the lack of definition. It would be even better to replace high- and low-income countries with other terms (e.g. higher/lower-income countries or countries with higher/lower income than the EU average). This would be especially necessary since the World Bank also uses the terms mentioned by the author (low- and high-income countries) but in a completely different way. According to this, low-income countries are those whose GNI per capita value is lower than 1085 dollars. The countries presented in the analysis all exceed this value without exception (they belong to the group of high-income countries).

In the last three sentences of the abstract, the author first writes about a positive relationship between the quality of life and research outputs, but then points out that the quality of life in lower-income countries has an inverse relationship with research performance. The contradiction does not necessarily exist between the two sentences, but by clarifying and supplementing the sentences, the reader could be helped to avoid misunderstanding the findings of the study. In general, the study explains several aspects of the phenomenon of the correlation between quality of life and research performance in lower-income countries, but at the same time, it would be important to make it even clearer and to emphasize that the research only examined high-income countries in which the standard of living is also relatively high (especially when compared to the poorest countries). In connection with future research, I consider it an interesting question to investigate if the low-income countries according to the above (World Bank) classification had also been included in the sample, what kind of correlation would have been found for the entire population and the individual groups in terms of quality of life and research performance.

All in all - in addition to considering some of the smaller suggestions above - I think the study is suitable for publication and I would like to congratulate the authors on their work! I wish you good luck!

Reviewer #2: I find this new, updated, revised version a much more matured, much more precise, much better defined material. Authors revised the introduction fundamentally in order to redefine the focuses of their paper, therefore the different levels affect scientific performance, moreover the spatial inequalities, differences in opportunities and incomes might help readers understand the differs in higher education potentials.

I would have only one suggestion / request you to use and embed this article into your paper that could also enhance the quality of your paper.

Császár, Z., Tarrosy, I., Dobos, G., Varjas, J., & Alpek, L. (2023). Changing geopolitics of higher education: economic effects of inbound international student mobility to Hungary. Journal of Geography in Higher Education, 47(2), 285–310. https://doi.org/10.1080/03098265.2022.2045572

7. PLOS authors have the option to publish the peer review history of their article (what does this mean?). If published, this will include your full peer review and any attached files.

Reviewer #1: No

Reviewer #2: No

---

## [Editor Report · Acceptance letter]

2 Jun 2024

PONE-D-23-26138R1 

PLOS ONE

Dear Dr. Credit, 

I'm pleased to inform you that your manuscript has been deemed suitable for publication in PLOS ONE. Congratulations! Your manuscript is now being handed over to our production team.

Kind regards, 

on behalf of

Dr. István Tarrósy 

Academic Editor

PLOS ONE